# The Impact of Perioperative Events on Cancer Recurrence and Metastasis in Patients after Radical Gastrectomy: A Review

**DOI:** 10.3390/cancers14143496

**Published:** 2022-07-19

**Authors:** Xing Zhi, Xiaohong Kuang, Jian Li

**Affiliations:** 1Department of General Surgery, Mianyang Central Hospital, School of Medicine, University of Electronic Science and Technology of China, Mianyang 621000, China; zhixing.my@126.com; 2Department of Hematology, The Third Hospital of Mianyang, Sichuan Mental Health Center, Mianyang 621000, China; kuangxiaohongvip@sina.com; 3Department of General Surgery, The Third Hospital of Mianyang, Sichuan Mental Health Center, Mianyang 621000, China

**Keywords:** gastric cancer, gastrectomy, recurrence, postoperative events, surgical stress response

## Abstract

**Simple Summary:**

Gastric cancer (GC) patients who are candidates for radical gastrectomy will experience various perioperative events, which have been shown to promote recurrence and decrease the long-term survival of GC patients. Therefore, although the perioperative period is relatively short, it is critical in determining the local recurrence and distant metastasis risk after radical gastrectomy. Herein, we will summarize the perioperative events and their effects on the long-term survival of patients with GC. Then, we discuss the possible mechanisms underlying perioperative vulnerability to cancer recurrence, directing the investigation of perioperative strategies to improve the survival of patients following gastrectomy.

**Abstract:**

Radical gastrectomy is a mainstay therapy for patients with locally resectable gastric cancer (GC). GC patients who are candidates for radical gastrectomy will experience at least part of the following perioperative events: surgery, anesthesia, pain, intraoperative blood loss, allogeneic blood transfusion, postoperative complications, and their related anxiety, depression and stress response. Considerable clinical studies have shown that these perioperative events can promote recurrence and decrease the long-term survival of GC patients. The mechanisms include activation of neural signaling and the inflammatory response, suppression of antimetastatic immunity, increased release of cancer cells into circulation, and delayed adjuvant therapy, which are involved in every step of the invasion-metastasis cascade. Having appreciated these perioperative events and their influence on the risk of GC recurrence, we can now use this knowledge to find strategies that might substantially prevent the deleterious recurrence-promoting effects of perioperative events, potentially increasing cancer-free survival in GC patients.

## 1. Introduction

Gastric cancer (GC) is the leading malignancy-related death worldwide, and radical gastrectomy is the foremost therapeutic strategy for patients with GC [1]. However, even though complete control of locoregional disease is considered to have been achieved, a considerable number of patients will experience tumor recurrence. GC patients who are candidates for radical gastrectomy will experience at least some of the following perioperative events: surgery, anesthesia, pain; intraoperative blood loss, allogeneic blood transfusion, and postoperative complications (POCs). Accumulating evidence suggests that the stress response to these surgery-related events will lead to the metastatic spread of many types of solid tumors, including GC [2]. For example, meta-analyses concluded that POCs were associated with worse survival, and allogeneic blood transfusion was shown to increase recurrence rates in GC patients following gastrectomy [3,4,5,6]. In addition, patients with GC undergoing extended surgery showed an increased risk of recurrence, especially peritoneal metastasis [7,8].

Therefore, although the perioperative period is relatively short, it is critical in determining the local recurrence and distant metastasis risk after radical gastrectomy. In this review, we aim to summarize the perioperative events and their effects on the long-term survival of patients with GC. Then, we discuss the possible mechanisms underlying perioperative vulnerability to cancer recurrence, directing the investigation of perioperative strategies to improve the survival of patients following gastrectomy.

## 2. Perioperative Events and Prognosis of GC Patients

### 2.1. Radical Gastrectomy

In the past four decades, several randomized controlled trials (RCTs) have been performed to compare the outcomes between surgical approaches with different extents in GC patients (Appendix A). The first two RCTs were conducted in Europe to determine the effectiveness of D2 lymphadenectomy, and both studies reported a high rate of POCs and surgery-related deaths but no additional survival benefits, however, after a median follow-up of 15 years, D2 lymphadenectomy is associated with lower locoregional recurrence and GC-related death rates than D1 surgery in Dutch D1D2 trial [7,9]. Other comparisons between different surgical extents, such as D2 vs. D2 plus para-aortic lymph node dissection (PAND), with vs. without splenectomy, bursectomy vs. nonbursectomy, and the left thoraco-abdominal (LTA) vs. transhiatal (TH) approach, were mainly carried out in East Asia [8,10,11,12,13]. All of them found that although most extended surgeries did not provide a survival advantage, they also had no significant negative effect on the recurrence of GC. In recent years, minimally invasive surgery (MIS) has dominated surgical oncology, including laparoscopic or robotic gastrectomy for early or advanced GC. However, only some postoperative recovery advantages have been demonstrated, and no survival benefit has been observed [14,15,16,17].

From these RCTs, we cannot conclude that more extensive surgery will lead to more GC recurrence. However, some points must be re-evaluated when interpreting these trials regarding the intensity of surgery. First, the probability of residual cancer cells is very low in early GC; therefore, when comparing the effects of different surgical approaches in patients with early GC, only differences in short-term outcomes can be detected, and the long-term survival will always be similar. Second, these trials aimed to determine the therapeutic efficacy of different surgical approaches but did not focus on the effects of surgery intensity. Therefore, individual trials only included specific GC subtypes; for example, only patients with cancer located in the gastric body or cardia with esophageal invasion of 3 cm or less were included in the JGOG0902 trial [8]. It is difficult to dissect the effects of surgery on the recurrence of cancer because radical surgery is still the main approach that can potentially cure a cancer patient. Extended surgery may indeed improve the survival of patients whose cancer cells have already spread distantly but are still within the scope of resection. Therefore, the therapeutic effects may offset its recurrence-promoting functions. Third, the majority of these studies were carried out in patients from East Asia, in whom aggressive gastrectomy for cancer can be easily performed due to their low body mass index, fewer comorbidities, and experienced surgical skills of surgeons resulting from the high incidence of GC. The differences in the intensity of the stress response to different surgeries may be smaller than we expected. Fourth, the majority of these studies were conducted before the establishment of scientific perioperative management, which has clearly reduced the stress response to surgery and improved the survival of patients in recent years. Therefore, all surgical approaches activated a strong stress response, and the differences in impacts on recurrence could not be detected. Fifth, in January 2007, the efficacy of adjuvant S-1 chemotherapy was demonstrated and established as the standardized treatment strategy in patients with stage II and III GC [18]. In trials performed after 2007, adjuvant chemotherapy was integrated into the perioperative management, which might have counteracted the survival impact of extended surgery.

Therefore, when we further analyzed these studies, some results indicated that extensive surgery might impact the survival of some patients with GC, although these analyses were *post hoc* and prone to bias. In the MRC trial, pancreatico-splenectomy was independently associated with poor survival when compared with both pancreas and spleen preservation [7]. Subgroup analysis found that less extensive surgery showed better survival in patients with more advanced diseases (T3 and T4, or node positive), which indicates that the probability of residual tumor cells is relatively high, and the negative effects of extended surgery on cancer recurrence can be revealed [10]. Therefore, when preoperative chemotherapy was given with the ability to minimize distant micrometastases, D2 plus PAND can provide better survival [19,20]. In the JCOG 0110 trial, the incidence of No. 10 metastasis was very low, and the majority of patients with No. 10 metastasis died of recurrence [11]. Several recent studies have investigated spleen-preserving hilar lymph node dissection, and nearly half of them concluded that prophylactic splenectomy to macroscopically remove splenic hilar negative lymph nodes will decrease the survival of GC patients when compared with spleen-preserving hilar lymph node dissection [21]. However, the majority of studies were retrospectively designed. In only one RCT, the 5-year survival rates were equal between the two groups. Furthermore, the 5-year survival rate of patients with No. 10 lymph node metastasis was 0 in both groups [13]. These findings indicate that when cancer cells are found in No. 10 lymph nodes, they are also disseminated into distant organs, splenectomy cannot cure the disease and may also promote the growth of pre-existing micrometastases, which is the case at least in patients with cancer not invading the great curvature. In addition, the differences in intensity may be the most obvious between LTA and TA gastrectomy; therefore, in the JCOG9502 trial, a nonsignificantly higher 5-year disease-free survival (DFS) rate was observed in the TH group, and a 22% lower 10-year overall survival (OS) rate and more peritoneal seeding were found in type 3 tumors treated by LTA [8]. This phenomenon has been validated in experimental studies, which have shown that not only a laparotomy, but also an isolated thoracotomy, might promote the seeding of colon cancer cells in the peritoneum, and thoraco-laparotomy results in a greater number of metastatic nodules and a shorter survival time in rats with intraperitoneal and intravenous inoculation of Sato lung cancer cells [22,23].

### 2.2. Anesthesia and/or Analgesia

Given that the majority of patients with GC will experience at least one anesthesia, the effects of different anesthetic approaches and drugs on the cancer outcomes of GC patients have been increasingly regarded as a research priority. The choices of anesthetic approach (general vs. neuraxial anesthesia), anesthetic agent (inhalational vs. intravenous), and accompanying prescribed analgesic drugs have long been assumed to influence the risk of cancer recurrence [24]. However, the available limited evidence regarding the effects of these techniques and drugs on cancer recurrence in GC has been inconclusive (Appendix A). Regarding anesthetic techniques, epidural anesthesia has populated in recent years in oncological surgery [24]. A small number of retrospective studies have reported the relationships between epidural anesthesia and GC patient outcomes with conflicting results. Only one study reported a positive effect of epidural anesthesia on OS but did not report the effects on cancer recurrence [25]. Another study reported that epidural anesthesia might decrease the 2-year recurrence of gastroesophageal cancer [26]. The other authors concluded that neither increased OS nor relapse-free survival (RFS) were observed in patients undergoing epidural anesthesia [27,28,29,30]. Three studies focused on the differences in cancer outcomes between propofol-based total intravenous anesthesia (TIVA) and inhalation anesthesia in patients with GC. Two studies reported an increased OS in the TIVA group, but the effects on cancer recurrence were not mentioned [31,32]. Another study reported a hazard ratio (HR) of 0.91 (0.50–1.67) for 1-year cancer-related mortality [33]. In addition, although preclinical studies have demonstrated that opioids can stimulate the progression of GC, their impact on cancer recurrence in clinical situations is unclear [34].

Therefore, the influence of anesthesia and/or analgesia on the prognosis of patients following gastrectomy remains debated. However, all of these studies were retrospectively designed with significant heterogeneity. The mechanisms through which different anesthetic techniques and drugs impact cancer recurrence were proposed to influence the stress response to surgery, which will last for several days to two weeks [35]. However, the efficacy and total dose-exposure to regional anesthesia have not always received much attention. In light of this, in the study by Hiller et al., longer administration and effective regional anesthesia improved the 2-year survival [26]. In addition, positive results tend to be observed in studies on gastrectomy with greater surgical intensity [31,33]. Despite these controversies, the cancer outcome effects should be taken into consideration when making a decision during the anesthesia of gastrectomy.

### 2.3. POCs

Gastrectomy is a technically demanding procedure, making POCs some of the most common perioperative events, reported to be in the range of 1.88–59.8% in a meta-analysis [3]. POCs were associated with an eventful postoperative recovery course and were also reported to increase cancer recurrence in GC patients following radical resection. Mounting evidence exists regarding the association between POCs and the recurrence or survival of patients with GC [3,36]. The majority of studies set OS as the primary endpoint and did not exclude early postoperative deaths, which may have diluted the prognostic influence of cancer recurrence. In addition, there was clear between-study heterogeneity regarding disease stage, surgical approach, and type and severity of POCs. Therefore, the conclusions were inconsistent. Despite these limitations, two meta-analyses concluded that POCs correlate with increased cancer recurrence (Appendix A) [3,36]. The negative effects remained when early deaths were excluded from the survival analysis [3]. These effects are mainly contributed by infectious and severe complications and are only found in advanced but not early GC. The underlying mechanisms are not completely clear, one possible explanation may be that POCs promote the progression of residual cancer cells mainly by exaggerating and prolonging the stress response initiated by gastrectomy. Therefore, minimizing micrometastases before surgery by neoadjuvant chemotherapy may abolish the recurrence-promoting role of POCs [37].

### 2.4. Anemia, Intraoperative Blood Loss and Blood Transfusion

Anemia is estimated to range from 27% to 44% preoperatively in GC patients, can also be found intraoperatively or postoperatively, and is associated with bleeding and the perioperative stress response [38]. Anemia leads to an eventful postoperative course as well as poor survival. Several retrospective studies and one meta-analysis have reported that pretreatment anemia was associated with poor disease-free survival (DFS); however, this was only the case in univariate analysis, indicating that anemia is not an independent predictive factor for cancer recurrence but an accompanying manifestation of other indicators with definite prognostic value, such as advanced stage and intraoperative bleeding (Appendix A) [38]. As mentioned above, D2 gastrectomy is technically demanding, and excessive intraoperative blood loss occurs repeatedly even at the hands of experienced surgeons. Most studies on intraoperative blood loss in gastrectomy have reported an increased recurrence rate (especially peritoneal metastasis) in patients with GC (Appendix A). Intraoperative blood loss inevitably increases the rates of perioperative allogeneic blood transfusion and POCs, which were also proposed to be negative prognostic factors. However, even after excluding these patients or adjusting for these confounding factors, the adverse effects of intraoperative blood loss on recurrence were still significant. Except for intraoperative blood loss, anemia also increased the need for perioperative transfusion. A number of studies and meta-analyses have been performed to analyze the association between allogeneic blood transfusion and recurrence in patients with GC. Although the results from individual studies were conflicting, three meta-analyses all concluded that allogeneic blood transfusion is negatively associated with cancer-related mortality and GC recurrence (Appendix A) [4,5,6].

### 2.5. Malnutrition and Nutritional Support

The pervasiveness of reduction of digestion and absorption, obstruction of the digestive tract, and anorexia caused by tumor-derived cytokines places patients with GC at high risk of malnutrition. Surgical trauma stress and eating restriction may further aggravate malnutrition. Therefore, nutrition screening and assessment is an important part of the perioperative management of patients with GC [39]. Numerous studies on the association between malnutrition and long-term survival in GC patients have been reported with various parameters to identify malnutrition, including food intake, body mass index, weight loss, body composition assessment, laboratory measures, or their combinations, such as the Global Leadership Initiative on Malnutrition (GLIM) criteria, prognostic nutritional index (PNI), and controlling nutritional status (CONUT). Not surprisingly, all of them reported a poor prognosis, including RFS, in GC patients with malnutrition [40,41,42]. As one of the few factors that can be corrected before surgery, numerous nutritional support formulas and approaches have been tested and compared during the perioperative period in patients with GC. However, all of them focused on short-term outcomes, such as POCs, immune function, and inflammatory response [39]. No results regarding the survival benefit of nutritional support have been reported. The reason may be that it is unethical to design an RCT with malnourished patients uncorrected. Given the encouraging short-term outcomes, perioperative nutritional support, especially some specific nutritional elements, should provide a survival advantage, which was validated in an RCT including patients with head and neck cancer [43].

## 3. Mechanisms Underlying the Increased Recurrence Rate Resulting from Perioperative Events

Radical gastrectomy is defined by no detectable tumors other than the primary lesion that can be removed by surgery. However, a considerable number of patients will experience recurrence following radical gastrectomy, indicating that these patients have tumor cells left after the surgery. These residual cancer cells may be pre-existing micrometastases, incompletely resected fractions of tumor cells or disseminated from the primary tumor in the operation. Whether these cancer cells grow to clinically apparent disease is determined by the balance between tumor surveillance of immune cells and the ability of tumor cells to survive, proliferate, and promote angiogenesis [44]. Following gastrectomy, the immune system is suppressed by various mechanisms and fails to eliminate these residual cancer cells, while the same mechanisms also support the survival, proliferation and angiogenesis of cancer cells, leading to local recurrence, peritoneal implantation or distant organ metastasis (Figure 1).

### 3.1. Perioperative Events Involved in Each Step of the Invasion-Metastasis Cascade

The invasion-metastasis cascade is a complicated but inefficient process. First, the cancer cells detach from the primary lesion and intravasate into the lymph or blood circulation, in which they must survive antitumor immunity and anoikis. Then, these cells extravasate into distant organs that have already been preconditioned to form a privileged microenvironment (premetastatic niche) [45]. In these places, whether colonized cancer cells become dormant or grow visible metastatic lesions is determined by perturbations of the local microenvironment and systemic functions, and all aspects can be affected by perioperative events [45]. The neural signaling and inflammation activated by perioperative events induce the epithelial-mesenchymal transition (EMT) of cancer cells, which is a well-established premise for cancer cells to acquire the ability to detach from primary lesions, migrate and invade adjacent tissues and lymphatic and/or blood circulation [46]. Perioperative platelet and neutrophil elevation caused by inflammation or splenectomy leads to the formation of microclots, ‘platelet cloaking’ and neutrophil extracellular traps (NETs), which afford protection to liberated circulating tumor cells (CTCs) from both vascular shear stress and detection by anticancer immune cells and facilitate extravasation to distant tissues [47,48]. Surgical stress responses, including inflammation and neural signaling, also denude the microcirculatory endothelium and attract mesenchymal stem cells to distant organs to create a premetastatic niche [49,50]. Locally or systematically activated inflammation also increases soluble growth factors, such as epidermal growth factor (EGF), platelet-derived growth factor (PDGF), and vascular endothelial growth factor (VEGF), thus activating dormant micrometastases, stimulating angiogenesis, inducing propagation and supporting the growth of residual or newly colonized cancer cells [51,52]. Throughout the invasion-metastasis cascade, immunosuppression caused by perioperative events also plays an important role in successfully forming a metastasis [35].

### 3.2. Physiological Responses to Perioperative Events

#### 3.2.1. Activation of Neural Signaling

The protumor roles of the sympathetic nervous system (SNS) and its released neurotransmitters have been validated in many types of cancer [53]. The abovementioned perioperative events, along with tissue damage during gastrectomy, pain and emotional changes, activate the hypothalamus pituitary adrenal (HPA) axis and SNS to release catecholamines into circulation [54]. Clinical studies have found that in GC patients undergoing more extended surgery or experiencing an eventful postoperative course, the levels of circulating catecholamines increase significantly [52]. The biological functions of catecholamines are mediated by adrenergic receptor (AR) families, with β2-AR being the main type expressed in GC [55]. The levels of noradrenaline and the expression of β2-AR were aberrantly elevated in GC tissues compared with adjacent normal gastric mucosa [55,56]. Catecholamines were shown to enhance the ability of primary GC cells to migrate to the liver and lung to form metastases, while propranolol can abolish the formation of metastatic lesions [55]. Higher protein levels of β2-AR were also associated with venous invasion, lymph node metastasis and poor prognosis in patients with GC [55,57,58]. The findings of several studies showed that activation of β2-AR promotes EMT and endows GC cells with stem cell-like properties through hypoxia inducible factor-1α (HIF-1α)-Snail, ERK, matrix metalloproteinases (MMPs) and STAT3-CD44 pathways, all of which are closely related to the invasion, migration, and dissemination of cancer [59,60,61,62]. In addition, lymphatic flow is also regulated by neural signaling, and high levels of circulating catecholamines have been found to increase flow through lymphatic vessels [63]. Given the high lymphatic vessel density in the stomach and its intraoperative rupture, the activation of neural signaling may promote the dissemination of cancer cells through lymphatic systems. Catecholamine signaling might also facilitate cancer metastasis by establishing a receptive environment for disseminated tumor cells (DTCs) [49]. Furthermore, catecholamines have been shown to suppress antimetastatic immunity directly by deactivating natural killer (NK) cells or cytotoxic T lymphocytes (CTLs) or indirectly by elevating the levels of immune inhibitory components, such as regulatory T (T_reg_) cells and M2 macrophages [64,65]. Together, these findings suggest that activation of neural signaling by perioperative events in GC patients is an important contributor to recurrence and metastasis.

#### 3.2.2. Activation of Inflammatory Responses

The survival of residual cancer cells after gastrectomy depends on several factors, of which inflammation contributes the most. Acute surgical stress activates the inflammatory response, while POCs further prolong its duration. The processes of wound healing after surgery share common inflammatory pathways with cancer evolution [66]. Therefore, several studies have reported that increased inflammatory indicators, such as the neutrophil to lymphocyte ratio (NLR), platelet to lymphocyte ratio (PLR), and C-reactive protein (CRP), are associated with cancer recurrence and poor prognosis following radical resection in many types of cancer, including GC [50,67,68]. When compared with levels before gastrectomy, inflammatory markers, including neutrophil percentage, CRP, procalcitonin (PCT) and plasma cortisol, were significantly increased after surgery [69]. These clinical findings indicate that gastrectomy and its associated events activate an inflammatory response, resulting in more recurrence after surgery. The inflammatory changes at the surgical site recruit various immune and inflammatory cells, which release humoral factors, such as VEGF and MMPs, all of which are important factors for the growth and dissemination of cancer cells [70]. Furthermore, the recruitment of fibroblasts and mesenchymal stem cells leads to growth factor secretion, and the systemic increased inflammatory reaction also creates a premetastatic niche, both of which provide a privileged environment for the growth of residual tumor cells and colonization by CTCs [51,71]. For example, interleukin-1 (IL-1) and tumor necrosis factor (TNF-α) were shown to enhance the adhesion of GC cells to mesothelial monolayers, which may partly account for peritoneal implants [72]. In addition, pre-existing dormant micrometastases can be awakened or propagated by local and systemic inflammatory responses. Gastrectomy, POCs, intraoperative blood loss and transfusion also increase the levels of prostaglandin E_2_ (PGE_2_) in the resection site and in circulation [73]. PGE_2_ facilitates the recurrence of cancer through actions on cancer cells directly or indirectly on immune cells. PGE_2_ plays an important role in GC metastasis by inducing sustained inflammation and promoting GC cell stemness, angiogenesis and invasion [74,75]. PGE_2_ has also been shown to decrease the number of activated CD8^+^ T cells, induce the expansion of T_reg_ cells, and convert T helper 1 (Th1) cell cytokines to cancer-promoting Th2 cytokines [76]. NK-cell infiltration was found to be negatively correlated with the expression level of cyclooxygenase-2 (COX-2), a rate-limiting enzyme for PGE_2_ synthesis, and NK-cell dysfunction was mainly induced by PGE_2_ derived from cultured GC cells [77].

#### 3.2.3. Suppression of Anticancer Immunity

Animal studies have provided unequivocal evidence in support of a role of the immune system in antimetastasis, which mainly relies on antimetastatic immune cells, including CTLs, NK cells, macrophages, and dendritic cells. Immunosuppression is an established phenomenon throughout the perioperative period that plays an important role in accelerating metastatic progression [35]. Multifactorial processes might lead to perioperative immunosuppression. The complex mechanisms through which these perioperative events weaken antimetastatic immunity are mainly activation of the abovementioned neural signaling and inflammatory responses, which have been reviewed elsewhere [2,35,73]. Postoperative immunosuppression has been observed in patients with GC. The total number of lymphocytes was found to significantly decrease after radical gastrectomy, which can last for more than 1 week, and immune checkpoint molecules, such as programmed cell death ligand 1 (PD-L1) and lymphocyte activation gene-3 (LAG-3), were upregulated on CD4+ and CD8+ T cells after surgery for GC [78]. Postoperative immunosuppression may facilitate the survival and growth of residual cancer cells, leading to detectable metastatic lesions.

### 3.3. Individual Event Aspects Affecting Recurrence

#### 3.3.1. Radical Gastrectomy

Gastrectomy itself and its related physiological perturbations (see Section 3.2) also have the probability of increasing the risk for cancer cell dissemination and progression of preexisting micrometastases. Manipulation and disruption of the tumor and its vasculature during surgery might release cancer cells into the circulation. CTC numbers have been found to increase following gastrectomy, and patients with postoperative increased CTCs have earlier and more hematogenous metastasis and shorter DFS [79]. DTCs in lymphatic vessels and sentinel lymph nodes have been observed in experimental animals and cancer patients [80,81]. Interstitial edema along with increased lymphatic clearance of cellular debris leads to lymphatic transit of residual cancer cells [82]. Transcoelomic dissemination of GC is another common route, and extended surgery has been shown to significantly increase the incidence of peritoneal recurrence [8,11,83]. Although studies on perioperative CTC changes and dissemination of cancer cells to the peritoneal cavity in patients undergoing gastrectomy are limited, it is possible that cancer cell dissemination during surgery contributes to peritoneal implants, distant metastasis and local recurrence.

#### 3.3.2. Anesthetic and Analgesic Drugs

Some anesthetic and analgesic drugs have an impact on GC recurrence. Although controversy exists and data are limited in GC, inhalational anesthetics, including isoflurane and sevoflurane, have been shown to promote cancer progression by immunosuppression and their cytoprotective effects [84,85]. Pain is an inevitable phenomenon during the perioperative period, especially after major surgeries, such as gastrectomy. Traditionally, opioid analgesics were widely used to relieve postoperative pain. No conclusive evidence suggests avoiding the use of opioids with the goal of reducing the risk of recurrence in GC, as no clinical evidence is available for GC, and the effect of opioids on the prognosis of patients with cancer types other than GC are inconclusive [86]. However, current preclinical evidence supports the direct role of stimulation of opioid receptors in survival, proliferation, migration, invasion, angiogenesis and cancer cells. Indirectly, opioids can exert stimulatory effects on cancer progression through the suppression of antimetastatic immune cells, including NK cells, CTLs, dendritic cells, and macrophages [87].

#### 3.3.3. POCs

The negative impacts of POCs on the survival of GC patients following radical resection have been supported by many studies [3,36]. Major POCs, which are defined as Clavien–Dindo (CD) III or higher by the majority of studies, need invasive intervention that will strengthen the surgical tress. POCs after gastrectomy, especially infectious complications, can actually prolong the postoperative inflammatory and immunosuppression processes, which can be measured by changes in daily temperature and WBC, CRP and NLR levels [88]. These effects were also validated in rodent models, surgery and intraabdominal sepsis, have been demonstrated to promote metastasis through systemic signaling, the expansion of regulatory T cells, reduction in the number of CD8 and NK cells, and the accumulation of tumor-associated macrophages and neutrophils [89]. Furthermore, lower rates of and delayed receipt of adjuvant therapy in patients who experience POCs is another proposed reason for their poor prognosis, which was supported by a retrospective analysis of the US GC Collaborative database [90]. The likelihood of patients who experienced POCs following gastrectomy to complete the adjuvant therapy was reduced by 50%. Therefore, among patients who received adjuvant and/or neoadjuvant therapy, the prognosis was similar between patients who did and did not experience a POC [89,91]. A mechanism to explain the prognostic benefit of neoadjuvant chemotherapy would be the effect of chemotherapy on microscopic residual disease. A preclinical rodent study demonstrated that isolated tumor cells and micrometastases are vulnerable to preoperative or postoperative fluoropyrimidine-based chemotherapy [92]. Therefore, it seems plausible that preoperative therapy may have treated the microscopic residual disease, so that it was less susceptible to activation and growth during POCs [89].

#### 3.3.4. Anemia, Hemorrhage and Blood Transfusion

Theoretically, anemia existing preoperatively or caused by blood loss intraoperatively will aggravate the hypoxic microenvironment. As an adaptive response to hypoxia, elevated expression of HIF-1 promotes the survival, migration, invasion and metastasis of cancer cells [93]. Despite causing anemia, intraoperative blood loss leads to increased cancer recurrence through several causes, including the spillage of cancer cells into the peritoneal cavity, immunosuppression of antitumor immunity, increased POCs and perioperative transfusion [94]. As a rapid and effective means to correct anemia and circulatory instability, allogeneic blood transfusion also has a negative impact on cancer-related outcomes mainly through the inhibition of host immunity and increasing the risk of POCs. These adverse effects are thought to mainly be caused by transfused allogenic leukocytes, however, current transfusion products are often leukodepleted, and no difference in the prognosis has also been found in comparative studies between leukocyte-depleted blood and non-leukocyte-depleted blood, indicating that the roles of these leukocytes remain unclear [95]. In addition, cell apoptosis occurring during refrigerated storage may also lead to immunosuppression [96]. Allogeneic blood transfusion induces several inflammatory mediators to inhibit IL-2 and interferon-γ (IFN-γ) production, suppress the function of anticancer cells, and increase immunosuppressive cells and PGs. Allogeneic blood transfusion also promotes invasion of cancer cells through IL-6, VEGF and hepatocyte growth factor (HGF) [95,96].

#### 3.3.5. Malnutrition and Nutritional Support

Currently, the underlying mechanisms by which malnutrition affects cancer recurrence remain uncertain. There was a significant correlation between malnutrition and poor patient-related or advanced tumor-related factors, such as older age, larger tumor size, and more advanced stages, which may increase the recurrence of GC [40,41,42]. Therefore, whether malnutrition is a cause or a consequence of GC remains unknown. However, preoperative malnutrition may interfere with treatment implementation for GC patients, even being unsuitable for adjuvant therapy in clinical practice, and was also significantly associated with the occurrence of POCs, suggesting that malnutrition can promote GC recurrence indirectly. Nutritional support can correct malnutrition and prevent these malnutrition-related disadvantages, along with its immune-enhancing function, thereby theoretically decreasing the recurrence of GC. However, excessive nutritional support, specifically parenteral nutrition, could potentially facilitate tumor-cell proliferation in preclinical studies [97]. Therefore, further investigations are needed to determine which patients should receive nutritional support and how to carry out nutritional support optimally.

## 4. Perspectives on Translating Therapy

As discussed above, the perioperative events contribute significantly to GC recurrence. Therefore, strategies should be explored and applied to minimize surgical stress and its contributing factors to potentially improve patients’ chances of surviving cancer-free. It is reasonable to choose surgical approaches based on the latest scientific evidence and avoid extended gastrectomy, which does not provide a survival advantage. Laparoscopic or robotic gastrectomy for early or even advanced GC, are increasingly used as alternatives to traditional open gastrectomy. Although no survival benefit has been observed, evidence for less blood loss, fewer POCs, and reduced surgical stress has been established [98,99]. Based on findings from RCTs, prophylactic splenectomy to remove macroscopically negative splenic hilar lymph nodes is not justified. Several retrospective studies suggest that spleen-preserving No. 10 lymph node dissection might be an alternative treatment for high-risk patients [21]. However, because patients with No. 10 lymph node metastasis will always experience recurrence, the survival benefit of spleen-preserving No. 10 lymph node dissection needs to be tested in future studies. From the perspective of operative experience, gastrectomy for GC should be performed by specialist surgeons in high-throughput centers to minimize POCs and improve prognosis [100]. The administration of preoperative chemotherapy to eliminate pre-existing micrometastases may be a possible way to prevent them from progressing to detectable lesions promoted by extensive surgical stress. Anti-adrenergic, anti-inflammatory, or antithrombotic drugs are all commonly applied in clinical practice and have been widely validated for their role in reducing surgical stress, which may also be applied in radical gastrectomy [101,102,103]. Until further evidence is obtained through dedicated clinical trials, when feasible propofol-based TIVA should be chosen with priority to inhalation anesthesia, and it seems preferable to replace opiates with epidural anesthesia/analgesia and/or NSAIDs for the suppression of postoperative pain and nociception [52,104]. In addition, patient blood management (PBM) has been examined in radical gastrectomy and was associated with a reduction in allogeneic blood transfusion rate and improvement in postoperative outcomes, as well as possible survival benefits [105,106]. Finally, along with the wide application of immunotherapy in clinical oncology and the opening of research on therapies targeting metastasis, how extrapolate these strategies to the perioperative period in GC patients deserves further investigation [35,107].

## 5. Conclusions

Radical gastrectomy is highly technically demanding, and patients undergoing this procedure are inevitably exposed to many perioperative events, such as anesthesia, analgesia, intraoperative blood loss, POCs, transfusion and psychological stress. Ample preclinical and clinical evidence suggests that perioperative events and their biological perturbations can promote metastatic progression and impact the long-term survival of patients with GC. The relationships between these events and GC recurrence are complex, and many of them remain inconclusive and poorly understood. The results of these studies do not imply that the current applied radical gastrectomy should be abandoned. In contrast, radical gastrectomy enables complete cure or long-term DFS of patients with locoregionally confined GC, the hard-won results obtained from painstaking sequential RCTs. However, having appreciated these perioperative events and their influence on the risk of GC recurrence and the benefits of many perioperative interventions, physicians can now use this knowledge to initiate urgently needed clinical trials to demonstrate their effects in GC and find inexpensive strategies that might substantially prevent the deleterious recurrence-promoting effects of perioperative events, potentially increasing cancer-free survival in GC patients.

## Figures and Tables

**Figure 1 cancers-14-03496-f001:**
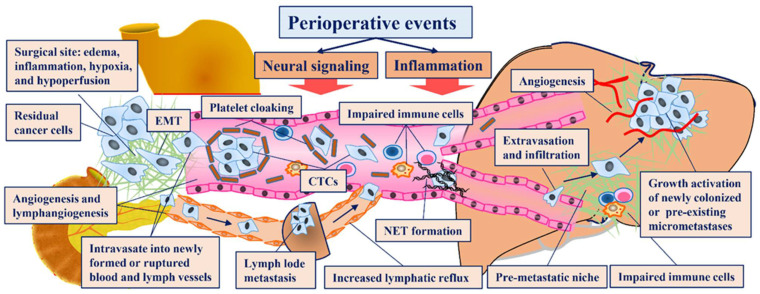
Putative mechanisms underlying recurrence and metastasis promoted by perioperative events in gastric cancer patients after radical gastrectomy. CTCs: Circulating tumor cells; EMT: Epithelial-mesenchymal transition; NETs: Neutrophil extracellular traps.

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
