# Peer review of "The Impact of Perioperative Events on Cancer Recurrence and Metastasis in Patients after Radical Gastrectomy: A Review"

_cancers, 2022, doi:10.3390/cancers14143496_

Round 1
Reviewer 1 Report
Review Title: The impact of perioperative events on cancer recurrence and metastasis in patients after radical gastrectomy. Cancers-1794417
Major:
The authors have written a profound review including recents scientific reports on perioperative events potentially influencing gastric cancer recurrence risk. However, I very much warn to use sentences like ..”opioid analgetics often increase recurrence rates” or “Based on currently available evidence, propofol-based TIVA should be chosen with priority to inhalation anesthesia.” There exists no profound scientific evidence supporting these hypotheses to my understanding.
For my understanding separating points under 3.3. from points under 2. seems quite artificial and does not lead to improved understanding.
Are there any animal studies using interventions perioperatively?
For an improved practical understanding, I suggest to include a table with only concluding recommendations like proven points (nutrition..) and negative .
Language is profound.
Minor:
Overall, I suggest to reduce some abbreviations (eg IBL or RFS – only twice used..).
Page 3 Line 99: I would suggest to exchange the verb “negate” as only persons can “negate” something.
Page 4 line 180: Statement of reason for promotion of cancer progression is speculative, I suggest to state this more carefully as there is no scientific proof.
Page 10 line 470: “textrapolate” might be “extrapolate”.
Author Response
Dear Reviewer:
Thank you for your letter and for the reviewers’ comments concerning our manuscript entitled “The impact of perioperative events on cancer recurrence and metastasis in patients after radical gastrectomy: A review” (Manuscript NO.: cancers-1794417). These comments are all valuable and very helpful. I have studied the comments carefully. After reading these comments carefully, we revised our manuscript. The main responses to the reviewer’s comments are as follows:
Major:
The authors have written a profound review including recents scientific reports on perioperative events potentially influencing gastric cancer recurrence risk. However, I very much warn to use sentences like ..”opioid analgetics often increase recurrence rates” or “Based on currently available evidence, propofol-based TIVA should be chosen with priority to inhalation anesthesia.” There exists no profound scientific evidence supporting these hypotheses to my understanding.
Response:
Thank you for your valuable comments. As many evidence included in this study were low levels, resulting from inherent limitations in designing such clinical trials as we discussed in our manuscript. We reviewed relevant literatures comprehensively, and still found that controversies exist, therefore, we changed the description way when we draw conclusions, which make them not too arbitrary but more suggestive, and some references were also updated to be more newly published.
- Regarding opioid analgetics, we revised the effects of opioid analgetics on GC only found in preclinical studies, evidence supports a clinical impact are lacking. Therefore, we added a comment as “No conclusive evidence to avoid the use of opioids with the goal of reducing the risk of recurrence in GC, as no clinical evidence is available for GC and the effect of opi-oids on the prognosis of patients with cancer types other than GC are inconclusive”.
- Regarding choose of anesthetic methods, we added “Until further evidence is obtained through dedicated clinical trials, when feasible” and cited two high impact papers to support our suggestion.
For my understanding separating points under 3.3. from points under 2. seems quite artificial and does not lead to improved understanding.
Response:
The problem of article format is a matter of different opinions. As your comments, these paragraphs are short and separation by subheadlines maybe not necessary. However, with my own experience, I always feel confused when I read a long section with paragraphs on various topics, which leads me to separate more in detail when I am writing a manuscript. In addition, such format can better echo the previous content. Therefore, I hope you may agree with us to keep them unchanged, thanks!
Are there any animal studies using interventions perioperatively?
Response:
We didn’t find any perioperatively interventional animal studies in GC. Several perioperative randomized controlled trials (RCTs) are ongoing, which test the effects of anti-adrenergic, anti-inflammatory, or antithrombotic drugs, and a few have already provided initial promising results, but none of them conducted in GC patients. Therefore, our suggestions on perioperative management of GC were mainly based on the prognostic studies on GC patients, basic researches and conclusions from RCTs on other cancer types. However, we did retrieved some animal studies on the mechanisms through which POCs to impact the recurrence of cancer, although not limit to GC cells, we cited the conclusions from them to support our discussion.
For an improved practical understanding, I suggest to include a table with only concluding recommendations like proven points (nutrition..) and negative .
Response:
Thank you for your suggestion. We have tried to summarize the underlying mechanisms and possible intervention recommendations in tables as you have suggested, however, as low level of and limited evidence in GC, we failed to complete this work. As mentioned above, although we have observed that many perioperative events may impact the recurrence of GC following radical resection, most of them are deduced from retrospectively designed prognostic studies, of which controversies also exist. In addition, relevant basic mechanistic and interventional clinical studies on GC are limited. Therefore, this manuscript is rather superficial, attributable to the ignorance of the negative effects of perioperative events of GC by the field. The purpose of our review is to throw a brick and attract jade, as we concluded that “having appreciated these perioperative events and their influence on the risk of GC recurrence and the benefits of many perioperative interventions, physicians can now use this knowledge to initiate urgently needed clinical trials to demonstrate their effects in GC and find inexpensive strategies that might substantially prevent the deleterious recurrence-promoting effects of perioperative events, potentially increasing cancer-free survival in GC patients”.
Minor:
Overall, I suggest to reduce some abbreviations (eg IBL or RFS – only twice used..).
Response:
Thank you for your comments. Following your suggesting, we have checked the manuscript and deleted some unconventional abbreviations, such as IBL and ABT.
Page 3 Line 99: I would suggest to exchange the verb “negate” as only persons can “negate” something.
Response:
As a non-native English speaker, some words maybe misused. Thank you for pointing out such misuse for us, and it was revised as “counteracted”.
Page 4 line 180: Statement of reason for promotion of cancer progression is speculative, I suggest to state this more carefully as there is no scientific proof.
Response:
As explained above, this manuscript is rather superficial and some statements are speculative. Therefore, following your suggestion, we revised the way of writing to avoid making an arbitrary conclusion. Such as the statement you comment, it was revised as “The underlying mechanisms are not completely clear, one possible ex-planation may be that POCs promote the progression of residual cancer cells mainly by exaggerating and prolonging the stress response initiated by gastrectomy.”
Page 10 line 470: “textrapolate” might be “extrapolate”.
Response:
We apologize for our careless and the mistake was corrected.

Reviewer 2 Report
This manuscript aims to review the recurrence-promoting effects of perioperative events after radical gastrectomy in gastric cancer patients. Furthermore, authors point out possible strategies to potentially improve gastric cancer (GC) prognosis.
Gastric surgery is a reality for many GC patients and the perioperative events mentioned in the paper are frequent in this population. The authors identified several changeable factors that could influence the prognosis of GC patients. Reviewing the impact in prognosis of these events will certainly help health teams to improve their work and the care of GC patients.
This is a well-structured review that looks into an important moment of cancer care. Initially, crucial factors that have shown worsening GC patients’ prognosis are pointed out. In a second part of the paper, it is detailed the pathophysiological and molecular mechanisms behind these events. Finally, the main conclusions are highlighted, as well as possible strategies and future areas of study with the purpose of improving the treatment of GC patients and their prognosis.
The text is mostly supported by pertinent references. However, in a few topics it would be important to review the supporting bibliography. In some cases, the citations presented don’t support the information written, in other cases it would be important to consider more recent data. Also, sometimes it would be pertinent to mention that studies cited were not designed to test the same hypothesis mentioned on the text (please see the Specific comments section).
The supplementary material is well organized and simplify the reading of the clinical trials results. Likewise, the image presented after topic 3 improves comprehension of the several mechanisms behind recurrence and metastasis promoted by perioperative events, reinforcing the information provided in text.
Regarding the perioperative events analyzed in this manuscript, they are relevant and frequent events on GC patients after gastrectomy. The information about each topic is concise and easy to interpret. Regarding neoadjuvant chemotherapy (NAC), it is briefly mentioned that it is modifying GC patients’ prognosis. There is also reference to the role it plays regarding micrometastasis and even an alert for the readers that studies cited in some topics were conducted before NAC be considered standard of care. Therefore, given the importance of this therapeutic approach and its establishment for a few years now, it would be interesting to explore a little bit more the influence of the NAC in the occurrence of these perioperative events and consequently its impact on GC prognosis.
Specific comments:
- Lines 44-46: please review the support bibliography; on the article referred as N. 4 the authors alert for the low quality of the evidence.
- Lines 46-48: reference N. 7 is from 1999 and reference N. 8 present data only about CG patients with esophageal invasion no other locations of GC. It should be reviewed if there is more recent evidence about this topic or a disclosure should be made in the text regarding these issues (old data and the other study included specifically patients with esophageal invasion).
- Lines 58-59: this is the first sentence of the topic “Radical gastrectomy” and the reference N.9 is about breast cancer patients. Despite the relevance of the information, the reader should be aware that the findings presented were not in GC patients.
- Lines 61-64: please complement this sentence. It is written that D2 lymphadenectomy showed no additional survival benefits, although it is important to notice that, once more, study N. 7 was published in 1999 (many things regarding surgery and treatment of GC patients as changed) and on the study cited as N.10 the authors concluded that “after a median follow-up of 15 years, D2 lymphadenectomy is associated with lower locoregional recurrence and gastric-cancer-related death rates than D1 surgery”, which isn’t according what was written in this manuscript.
- Lines 126 -128: citation N. 23 is from 2005 and refers to colorectal cancer; citation N.24 dates from 1980. Please check if there is more recent data about this issue.
- Lines 416 - 417: Please clarify this sentence. It is mentioned that adverse effects mentioned may be caused by transfused allogenic leukocytes. However, nowadays most allogenic blood transfusions are leukodepleted. Also, in reference N.95 it is mentioned that “no difference in the prognosis was found in comparative studies between leukocyte-depleted blood and non-leukocyte-depleted blood".

Author Response
Dear Reviewer:
Thank you for your letter and for the reviewers’ comments concerning our manuscript entitled “The impact of perioperative events on cancer recurrence and metastasis in patients after radical gastrectomy: A review” (Manuscript NO.: cancers-1794417). These comments are all valuable and very helpful. I have studied the comments carefully. After reading these comments carefully, we revised our manuscript. The main responses to the reviewer’s comments are as follows:
Therefore, given the importance of this therapeutic approach and its establishment for a few years now, it would be interesting to explore a little bit more the influence of the NAC in the occurrence of these perioperative events and consequently its impact on GC prognosis.
Response:
Thank you for your comments. So far as we know, only three papers have devoted to investigate the effects of (neo)adjuvant chemotherapy on survival of GC patients with postoperative complications, and an animal studies have been retrieved. Therefore, follow your constructive suggestions, we expanded the discussion based on these findings.
Specific comments:
Lines 44-46: please review the support bibliography; on the article referred as N. 4 the authors alert for the low quality of the evidence.
Response:
Thank you for your comments. As many evidence included in this study were retrospectively designed with low levels, especially for transfusion and postoperative complications, for which it isn’t feasible to validate them by RCTs. Regarding transfusion, after comprehensively searching, we found three meta-analysis, although inherent limitations are inevitable, they did support the negative effects of transfusion on recurrence of GC via various statistical methods to adjust confounding factors. Therefore, if we plan to discuss this topic, these bibliographies may be available evidence with the highest levels. So, please allow us to keep such citations.
Lines 46-48: reference N. 7 is from 1999 and reference N. 8 present data only about CG patients with esophageal invasion no other locations of GC. It should be reviewed if there is more recent evidence about this topic or a disclosure should be made in the text regarding these issues (old data and the other study included specifically patients with esophageal invasion).
Response:
As no clinical trials have been designed to validate the effects of surgical intensity on recurrence of GC, speculation can only be obtained from trials comparing two surgical methods with different intensity. Therefore, we systemically analyzed such clinical trials, and was limited to RCTs, and we found these two trials did reveal a worse survival in patients underwent an extended gastrectomy. Although majority of RCTs on surgical methods in GC were conducted two decades ago, their findings established the fundamental evidence of current standardized gastrectomy for GC, such as D2, no PAND, and no splenectomy. Therefore, no more recent such evidence are available.
Nevertheless, these trials do within the aim of our discussion, ie. more extended surgery may increase the recurrence of GC, no matter when they are conducted, we here only need an evidence. However, as your concerning, we have some disclosures in the text regarding these issues. Such as, “individual trials only included specific GC subtypes; for example, only patients with cancer located in the gastric body or cardia with esophageal invasion of 3 cm or less were included in the JGOG0902 trial” “…, which is the case at least in patients with cancer not invading the great curvature.”
Lines 58-59: this is the first sentence of the topic “Radical gastrectomy” and the reference N.9 is about breast cancer patients. Despite the relevance of the information, the reader should be aware that the findings presented were not in GC patients.
Response:
Thank you for your comments. As no clinical trials have been designed to validate the effects of surgical intensity on recurrence of GC, our intention is to lead to the discussion of GC through relevant conclusions in other cancer types. For avoiding misinterpretation as your concerning, we decided to delete this sentence.
Lines 61-64: please complement this sentence. It is written that D2 lymphadenectomy showed no additional survival benefits, although it is important to notice that, once more, study N. 7 was published in 1999 (many things regarding surgery and treatment of GC patients as changed) and on the study cited as N.10 the authors concluded that “after a median follow-up of 15 years, D2 lymphadenectomy is associated with lower locoregional recurrence and gastric-cancer-related death rates than D1 surgery”, which isn’t according what was written in this manuscript.
Response:
Thank you for your comments. Here we have made some misinterpretation. Therefore, we added this conclusion to our manuscript.
Lines 126 -128: citation N. 23 is from 2005 and refers to colorectal cancer; citation N.24 dates from 1980. Please check if there is more recent data about this issue.
Response:
Thank you for your comments. So far as we know, they are the only two animal studies to evaluate the effects of surgery relevant to GC on peritoneal implantation and lung metastasis, although colon and lung cancer cell line were used in them. To be more specific, we clearly indicated it in our manuscript.
Lines 416 - 417: Please clarify this sentence. It is mentioned that adverse effects mentioned may be caused by transfused allogenic leukocytes. However, nowadays most allogenic blood transfusions are leukodepleted. Also, in reference N.95 it is mentioned that “no difference in the prognosis was found in comparative studies between leukocyte-depleted blood and non-leukocyte-depleted blood".
Response:
Here we did make an arbitrary conclusion. As you have commented, controversies exist regarding the effects of leukocyte in transfused blood on the prognosis of patients with cancer. Therefore, we revised the sentence as “These adverse effects are thought to mainly be caused by transfused allogenic leuko-cytes, however, current transfusion products are often leukodepleted, and no difference in the prognosis has also been found in comparative studies between leukocyte-depleted blood and non-leukocyte-depleted blood, indicating that the roles of these leukocytes remain unclear”.

Round 2
Reviewer 1 Report
I agree with all answers.